# The Role of Metals in the Neuroregenerative Action of BDNF, GDNF, NGF and Other Neurotrophic Factors

**DOI:** 10.3390/biom12081015

**Published:** 2022-07-22

**Authors:** Vincenzo Giuseppe Nicoletti, Krisztián Pajer, Damiano Calcagno, Gholam Pajenda, Antal Nógrádi

**Affiliations:** 1Department of Biomedical and Biotechnological Sciences (BIOMETEC), Section of Medical Biochemistry, University of Catania, 95124 Catania, Italy; nicovigi@unict.it (V.G.N.); liberiano@hotmail.it (D.C.); 2Department of Anatomy, Histology and Embryology, Albert Szent-Györgyi Medical School, University of Szeged, 6720 Szeged, Hungary; pajer.krisztian@med.u-szeged.hu; 3Ludwig Boltzmann Institute for Experimental and Clinical Traumatology, Research Centre for Traumatology of the Austrian Workers, 1200 Vienna, Austria; gholam.pajenda@meduniwien.ac.at; 4Department for Trauma Surgery, Medical University Vienna, 1090 Vienna, Austria

**Keywords:** BDNF, GDNF, neurotrophic factors, metal ions, neural regeneration

## Abstract

Mature neurotrophic factors and their propeptides play key roles ranging from the regulation of neuronal growth and differentiation to prominent participation in neuronal survival and recovery after injury. Their signaling pathways sculpture neuronal circuits during brain development and regulate adaptive neuroplasticity. In addition, neurotrophic factors provide trophic support for damaged neurons, giving them a greater capacity to survive and maintain their potential to regenerate their axons. Therefore, the modulation of these factors can be a valuable target for treating or preventing neurologic disorders and age-dependent cognitive decline. Neuroregenerative medicine can take great advantage by the deepening of our knowledge on the molecular mechanisms underlying the properties of neurotrophic factors. It is indeed an intriguing topic that a significant interplay between neurotrophic factors and various metals can modulate the outcome of neuronal recovery. This review is particularly focused on the roles of GDNF, BDNF and NGF in motoneuron survival and recovery from injuries and evaluates the therapeutic potential of various neurotrophic factors in neuronal regeneration. The key role of metal homeostasis/dyshomeostasis and metal interaction with neurotrophic factors on neuronal pathophysiology is also highlighted as a novel mechanism and potential target for neuronal recovery. The progress in mechanistic studies in the field of neurotrophic factor-mediated neuroprotection and neural regeneration, aiming at a complete understanding of integrated pathways, offers possibilities for the development of novel neuroregenerative therapeutic approaches.

## 1. Introduction

Neurotrophin (NT) modulation can be a valuable target for treating or preventing neurologic disorders and age-dependent cognitive decline. Neuroregenerative medicine can take particular advantage through the deepening of knowledge of the molecular mechanisms underlying the NT properties. It is indeed intriguing that a significant interplay between NTs and essential metals can modulate the neuronal recovery outcome.

Motoneuron recovery represents a well-known example in the studies of neuroregeneration. Indeed, motoneuron degeneration and regenerative outcome received large attention, owing to their implications for muscular control and skilled activities. Remarkably, in contrast to CNS, motor neurons are autonomously able, under certain conditions, to support robust regenerative responses after peripheral nerve injuries [1].

This feature makes motoneuron recovery from insults a compelling model to study the molecular events that govern neuroregenerative pathways. On the contrary, regeneration after severe acute trauma or chronic neurodegeneration from underlying neurodegenerative disease is often irreversible with a profound impact on neuromuscular functions. Thus, amyotrophic lateral sclerosis (ALS), a disease that shows selective motor neuron degeneration with typical signs including axonal damage, mitochondrial dysfunction, cytoskeletal deterioration, and muscle denervation, gave a further model system for studies on motoneurons pathophysiology [2].

This review aims to emphasize the relevant role of NTs, particularly GDNF, BDNF, and NGF, in the process of neural recovery and motoneuron regeneration, being able to protect motoneurons from various insults and support restoration of lost neuromuscular interplay. NTs have been widely explored as therapeutic targets in different experimental models of motor neuron damage and ALS, and also in age-related neurodegenerative diseases. This review will also evaluate the key role of metal homeostasis/dyshomeostasis and metal interactivity with NTs on neuronal pathophysiology. This is a new scheme and potential target for neurorecovery that suggests further strategies and tools to deepen the knowledge on multiple NT activities.

## 2. Motoneuron Injury and Degeneration

Injury close to the cell body of motoneurons results in the death of the vast majority of injured motoneurons [3,4,5]. The morphological changes of the damaged motoneurons are initiated within a few days following ventral root avulsion injury. The cell body of the motoneurons becomes swollen and the nucleus migrates toward the periphery of cells. The Nissl bodies (tigroid) break up into several small units and appear to dissolve, which is a process known as chromatolysis [6,7]. Injury to the ventral root induces biochemical cascades in spinal motoneurons, resulting in glutamate-mediated excitotoxic events [8]. Injury upregulates the subtypes of NMDA and AMPA/kainate receptors which play a key role in glutamate-mediated excitotoxicity [9]. Activation of NMDA and AMPA receptors leads to the influx of Ca^2+^ ions into neurons and the increased cytoplasmatic concentration of Ca^2+^ induces a cascade of secondary processes, resulting in cell death. Glutamate release and thus excitotoxicity can be effectively blocked by presynaptic glutamate release inhibitors—for example, riluzole (2-amino-6-trifluoromethoxybenzothiazole) [10].

Koliatsos et al. [3] have demonstrated that avulsion injuries result in the retrograde death of the majority of the affected motoneurons in the first 2 weeks after injury. The outcome of motoneuron damage depends on the distance of the implied lesion from the cell body of the injured motoneurons. If the axotomy is performed close to the perikaryon most of the affected motoneurons die. On the other hand, motoneurons survive after axotomy of a peripheral nerve, if at least a 5 mm segment of the nerve remains proximal to the injury site and thus provides trophic support for the damaged motoneurons [11]. If the avulsed ventral root is surgically re-implanted into the spinal cord segment soon after injury, some of the injured motoneurons survive and maintain their capacity for axonal regrowth [5,10,11]. Cervical motoneurons have a greater capacity to survive and re-innervate target muscles than motoneurons localized in the lumbar spinal cord segments. This phenomenon is likely due to (a) the shorter distance that axons of the surviving neurons have to bridge to reach their targets and (b) the fact that the distal degenerated peripheral nerve segments may undergo “predegeneration”; an effect considerably slowing down or prohibiting the growth of regenerating axons [12,13,14]. It has been earlier recognized that successful reinnervation may occur only if a satisfactory number of surviving motoneurons remains in the spinal cord after avulsion injury. Therefore, different strategies to prevent the death of the injured motoneurons have been developed in the last two decades including various growth factors [15,16,17].

Neurotrophic factors are essential proteins for neuronal differentiation during development [18] and they also support neuronal maintenance in the adult central nervous system (CNS), modulate neuronal plasticity and synaptic transmission [13]. The neurotrophic factor family has several members and many of these have more or less neuroregenerative effects. An expanded repertoire of neurotrophic factors is now available for improving the outcome of motoneuron injury, especially the administration of brain- or glial cell line-derived neurotrophic factors (BDNF and GDNF, respectively).

## 3. Molecular Pathways of BDNF GDNF and NGF

Brain-derived neurotrophic factor BDNF was described in the 1980s by Barde et al. and belongs to the family of neurotrophins [19]. BDNF acts on certain neurons, helping to support the survival and differentiation of distinct neurons and synaptic plasticity in the CNS [18,20]. Its active form can be found in various parts of the brain and is also expressed in the retina, motoneurons, and skeletal muscle [21,22,23]. BDNF is first synthesized as a precursor, pro-BDNF in the endoplasmic reticulum, that acts as a biologically active factor, different from mature BDNF. Pro-BDNF can reduce dendritic complexity and synaptic plasticity in the hippocampus, while mature BDNF exhibits an opposing function in the CNS [24,25]. Pro-BDNF predominates in developmental stages, whereas mature BDNF is expressed in considerable amounts in adulthood [25].

BDNF mediates its effect through the TrkB (pronounced “Track B”) and the pan-neurotrophin receptor p75NTR. The latter was initially identified as the Low-affinity Nerve Growth Factor Receptor (LNGFR) and belongs to the tumor necrosis factor (TNF) receptor family [26]. TrkB receptors are present both on the presynaptic and postsynaptic membranes and are ~145 kDa membrane-bound glycoproteins. They bind with high affinity to BDNF, whereupon they dimerize and transphosphorylate each other. These post-translational modifications further induce phosphorylation of other tyrosine residues that operate as a docking site for adaptor proteins which launch further intracellular signaling pathways [18]. The cascades activate protein kinase B (AKT) and mitogen-activated protein kinase (MAPK) pathways that regulate several gene expression processes via transcription factor cAMP response element binding protein (CREB) and modulate apoptotic, survival mechanism and mammalian target of rapamycin (mTor) [27,28].

GDNF was first described as a neurotrophic factor in 1991 by Engele et al. [29]. GDNF is mainly expressed by neurons in the developing and adult CNS but is released from glial cells, too [30,31]. Numerous studies have shown that GDNF promotes differentiation of neurons, prevents apoptosis and enhances survival of injured motoneurons induced by ventral root avulsion injury [15,16]. GDNF preferentially forms a complex with the GDNF family receptor α1 (GFR α1, a glycosylphosphatidylinositol [GPI]-linked cell surface receptor), which is anchored to the plasma membrane of neurons. The GDNF-GFR α1 complex interacts with a receptor tyrosine kinase, RET receptors, resulting in activation of the intracellular kinase domain to induce multiple intracellular signaling pathways [32]. RET is mainly expressed in sensory neurons, dopaminergic and noradrenergic neurons and plays an essential role in the development of the sympathetic, parasympathetic and enteric nervous systems [33,34,35]. Activation of RET receptors induces various signaling pathways such as MAPK, phosphoinositide 3-kinase, and the phosphoinositide phospholipase C-γ pathway, which regulate cell survival, differentiation, proliferation, migration, neurite outgrowth, and synaptic plasticity [36].

Nerve growth factor (NGF) was first described in chicken embryo by Levi-Montalcini and Hamburger [37]. NGF binds to the tropomyosin receptor kinase A (trkA) and the p75 neurotrophin receptor (p75NTR). TrkA has a high affinity for NGF. The NGF–trkA interaction activates various molecular pathways including the phospholipase C-γ (PLCγ), the mitogen-activated protein kinase (MAPK)/Erk and the phosphoinositide 3-kinase (PI3K) pathways [38]. While NGF has affinity to both receptors, p75 NTR shows higher affinity for pro-NGF than NGF itself. p75NTR is a Trk co-receptor that can activate signaling cascades such as the NF-κB, Akt, and JNK pathways, resulting in the induction of apoptosis or in the promotion of survival of neurons [39]. As far as we are concerned, extraocular motoneurons are sensitive and responsive to NGF treatment [40].

## 4. Therapeutic Potential of BDNF, GDNF and NGF following Motoneuron Injury

Most of the studies concerning motoneuron survival and regeneration have focused on the neuroprotective role of different neurotrophic factors, such as BDNF and GDNF, known to promote neuronal survival and axonal growth both in vitro and in vivo [15,16,41]. These factors have been shown to induce extensive survival of neurons and partial or aberrant axonal regeneration after injuries, but the efficacy of these factors depends on their continuous supply. The side effect of this uncontrolled and persistent expression of neurotrophic factors at the site of injury was the formation of irregular axon coils, likely due to the trapping effect of neurotrophic factors on the injured axons [42]. A study by Novikov et al. demonstrated that exogenous treatment with BDNF rescued axotomized spinal motoneurons and also promoted partial recovery of the afferents [38]. However, long-term infusion of BDNF has other disadvantages, too, e.g., loss of S-type boutons on the injured and intact motoneurons [42,43]. A serious side effect of intraspinal axonal sprouting induced by BDNF treatment is spasticity, which may seriously jeopardize the positive therapeutic effects [44]. One of the promising methods to overcome these problems is the use of transplanted cells with neurotrophic factor expression capacity for more localized and regulated effects. Different groups have provided evidence that BDNF-expressing cells support extensive axonal growth at sites of spinal cord injury [15,45].

Eggers et al. created gradients of GDNF in the sciatic nerve 2 weeks after a ventral root avulsion (Figure 1A,B) [41]. The gradually increasing lentiviral-mediated GDNF expression results in increased axon numbers and formation of nerve coils. Moreover, the density of Schwann cells and motor axon sprouting were increased. The continuous expression of GDNF did not enhance the survival of motoneurons or their regeneration [41].

In other approaches, a viral or non-viral vector system was used to assess the efficacy of time-restricted BDNF and GDNF expression on motor neuron survival and axon regeneration following ventral root injury [15,46]. Time-restricted expression of GDNF hinders the axon coil formation and a considerable number of regenerating axons showed elongative growth pattern (Figure 1C) [16]. The time-restricted expression of GDNF enhanced the motoneuron survival in the lumbar segment and their axons reinnervated the nerve. The temporally expressed GDNF enhanced motoneuron survival in cervical segments following ventral root avulsion injury and promoted axon regeneration and muscle reinnervation. The expression of BDNF was capable of itself to induce as high amount of motoneuron survival and regeneration as individual production of GDNF (Figure 1D) [15]. However, it is important to note that the combined action of BDNF and GDNF had no synergistic effect on the regeneration of injured motoneurons and did not induce significantly improved functional recovery [15]. Further studies are required to be performed to investigate the time-restricted sequential expression effect of BDNF and GDNF on motoneuron survival and regeneration.

NGF is also a promising neurotrophic factor that is mainly used to induce nerve regeneration following peripheral nerve injury. Kemp and his colleagues have shown in an elegant study that NGF has a bell-shaped dose response curve for axonal regeneration. The optimal dose of NGF is 80 ng/day for 3 weeks that induced the best sensorimotor recovery compared to all other treatment groups [47].

To reach the potential neuroregenerative effect of NGF, NGF-loaded delivery systems have been used to enhance axon regeneration following peripheral nerve injury. These nerve conduits provide a suitable alternative to autologous nerve grafts and can release NGF for extended periods of time [48]. Several studies have reported that nerve conduits treated by NGF promote peripheral nerve regeneration efficiently [49,50,51]. A longitudinally oriented collagen conduit combined with NGF was able to reconstruct a long distance (35 mm) nerve gap via axonal regeneration following sciatic nerve injury in dog [50]. Similarly, thermo-sensitive heparin-poloxamer hydrogel co-delivered with NGF facilitated Schwann cell proliferation, enhanced axonal regeneration and remyelination, and improved recovery of motor function following sciatic nerve crush injury in diabetic rats [52]. Enzymatically cross-linked silk fibroin-based conduits were also used as a platform for the controlled delivery of NGF [53]. In another experimental set up NGF was loaded in the size-tunable microfluidic hollow fibers, which was released gradually and promoted a rapid morphological and functional axon regeneration in rats with 5-mm sciatic nerve defects [54]. All together, these results show that NGF loaded in nerve conduit possesses the capacity of promoting nerve regeneration after peripheral nerve injury; however, further studies are required to achieve the best and the perfectly safe therapeutic approach.

## 5. Neurotrophins and Metals

In the last two decades, it has emerged that there is a strict interplay between NTs and various metals. Metals can induce conformational changes by direct interaction with NTs, hence influencing the receptor recognition processes, or can contribute to their expression/secretion [55]. Intricate mechanisms of metallostasis have a critical impact on the NT activities in supporting neuronal physiology and also play a significant role in pathological conditions [56,57] (Figure 2 and Table 1).

Environmental metal neurotoxicity is not addressed in this review; however, considerable evidence suggests that targeting the metal homeostasis represents a new challenge in the exploration of alternative ways to gain neuronal functional recovery [58,59,60]. While these studies do not specifically target motor neurons, they can provide useful information to better understand motor neuron physiology and discover new therapeutic approaches for neural recovery.

The activity of NTs can be modulated differently by copper and zinc, two transition metals highly implicated in neuronal physiology and pathology. Intriguingly, their direct interaction with these metals can exert different and sometimes even opposite effects. In neuronal cell cultures, Zn^2+^ treatment and increased binding to BDNF have been associated with proliferative activity. Conversely, the effect of Cu^2+^ addition on the BDNF activity is the opposite. Interestingly, the effects of these metal ions are reversed in the case of cell culture treatment with NGF: the presence of Cu^2+^ exerts proliferative activity, whereas cell proliferation is repressed after Zn^2+^ addition [61,62]. Noteworthy, in vitro evidence indicates that high concentrations of Zn^2+^ and Cu^2+^ can produce conformational changes of different NTs and, in turn, ineffective interaction with specific receptors that affect trophic and antioxidant properties with consequent detrimental effect on motoneuron viability [63], or blocking of the NGF-mediated neurite outgrowth [64,65]. However, Zn^2+^ has been indicated as a key cofactor for the protective activity of NGF, that could be related to either metals inhibition of p75-mediated apoptotic cascades, or metals participation in neurotrophic signaling via the TrkA receptor [66,67,68]. Although the signaling induced by NTs is well characterized, ligand-independent activation of TrkB receptor has been also observed as solely mediated by zinc [69,70]. Zinc promotes a signaling event critical for transactivation of TrkB by preferential phosphorylation of Tyr-705/Tyr-706 of TrkB by a Src family kinase (SFK)-dependent, but TrkB kinase-independent mechanism, that implicates a regulatory role of SFKs in TrkB activation by BDNF [71]. Similarly, zinc induces transactivation of the EGF receptor (EGFR) by Src-dependent phosphorylation of Tyr-845 of EGFR [72]. These events are of particular relevance to neural activity involved in long-term potentiation (LTP), mediated by the release of zinc stored in secretory vesicles [73]. In this respect, it is worth mentioning the metal ions effects on glutamatergic synapses in the hippocampus, especially considering that this brain area is involved in learning and memory and is the brain area with the highest levels of Zn^2+^ and Cu^2+^, and NT activity [74,75]. Other evidence supports the hypothesis that also released Cu^2+^ ions can transactivate TrkB through extracellular signal-regulated kinase 1/2 and Src tyrosine kinase. This Cu-mediated pathway seems associated with increased activity of matrix metalloproteinase 2 and 9, which contributes to increased secretion of pro-BDNF and mature BDNF in cortical neurons and during wound healing [76,77].

A bidirectional interplay between metals and NTs should be also taken into consideration. BDNF and GDNF, as well as other factors including CNTF (Ciliary neurotrophic factor) and PEDF (pigment epithelium-derived factor), have been shown to modulate the expression of zinc(II) influx transporter, ZIP2, which increases metal ion uptake and intracellular Zn^2+^ level in RPE (retinal pigment epithelium) cells. However, differential effects on other zinc transporters are observed with other NTs: CNTF downregulates the expression of ZIP4, ZIP14, and ZnT6; PEDF downregulates ZIP4 and ZIP14; GDNF acts similarly on ZnT6; both PEDF and GDNF promote higher levels of ZnT2; PEDF upregulates the expression of ZnT3, the primary zinc loader in the brain [78].

Various NTs, including BDNF, NGF, and GDNF, as well as their receptors can also be upregulated by lithium, a metal with a long-lasting history of use as first-line drug for treating bipolar disorder and depression [79,80]. This effect explains the neuroprotective properties of lithium against injuries and in neuroregeneration.

On account of the aforementioned points, it is evident that transition metal–NT interactions could trigger various signaling pathways, suggesting that any intervention on transition metals homeostasis can produce relevant effects not restricted to various metabolic processes, including enzyme activities and gene expression, but nervous system physiology and many pathologic statuses (e.g., Alzheimer’s diseases) can be strongly challenged as well [81,82]. This issue is still debated but could deserve deeper understanding to highlight specific pathways and cell-specific targeting. This can open the way to gain valuable therapeutic tools for tissue-directed neuronal protection and recovery.

**Table 1 biomolecules-12-01015-t001:** Neurotrophin–metal interplay and metal-related neuromodulatory effects.

NTs	Metal	Mechanism	Effect	Ref.
n.a.	Various	Metal dysregulation	Motoneuron pathology	[83,84,85,86]
Various	Copper or Zinc	Protein conformational changes	Protein-misfolding diseases	[58,59,60,87,88]
NGF	Copper or Zinc	Neurotrophic effects	Neuronal cell culture changes in proliferation	[62]
BDNF				[63]
BDNF	Copper or Zinc	Direct interaction and NT conformational changes	Altered motoneuron trophic signaling	[63]
NGF			Altered neuronal trophic signaling	[64,65]
NGF	Zinc	Direct interaction and receptor transactivation	Neuroprotective outcome	[66,67,68]
BDNF	Zinc	Receptor transactivation	Tyr phosphorylation cascade (SFK)	[69,70,71]
EGF				[60,72]
BDNF	Copper	NT level changes	Increased secretion of pro-BDNF and mature BDNF	[76,77]
BDNF	Zinc	NT-mediated changes of metal homeostasis	Modulation of zinc transporters	[78]
GDNF				
Other NTs				
BDNF, GDNF, NGF	Lithium	NT upregulation	Neuroprotection, neuroregeneration, and axons remyelination	[79,80,89,90]
BDNF	Zinc			[91,92]
n.a.	Lithium	Others	Antidepressant	[86,89,93,94]
		Induction of autophagy	protection from spinal cord injury	[90,95,96,97,98]

n.a. = not applicable.

Table 1 shows that a functional bidirectional interplay between NTs and essential metals is emerging as a relevant mechanism in the processes that sustain neuronal protection and recovery from various injuries. NT levels can be modulated by the presence of various metals, but NTs can also modulate metals homeostasis, and metals per se can also trigger neurotrophic signaling. However, metal dyshomeostasis is known to produce neurotoxic conditions as well.

## 6. Current Insight into the Potential of Regenerative Medicine Combining Metal Modulation and Neurotrophins

Considerable efforts have been dedicated to the search for effective treatments of injured neurons either during a degenerative disease or after traumatic events. The study of the role of NTs in the recovery of nerve function and nerve regeneration is one the most important issues to achieve this goal. Many growth factors and signaling pathways are known to be involved to various extents in axonal maturation as well as restoration after injury [82,99]. Noteworthy, emerging evidence highlights the role of metals in NT upregulation or modulation of specific signaling cascades and provides new perspectives in the search for advanced therapeutic tools. The importance of metals for neuronal pathophysiology is currently mainly studied in the function of neurodegenerative diseases. The molecular events described to date, however, could also be of interest to understand the molecular pathways involved in motoneuron recovery, and this review highlights the need to expand studies in particular on neurotrophin–metal correlations.

Despite conflicting results that have been reported so far, the study of the mimetic activity of NGF peptides showed that copper and zinc ions exert modulatory effects through conformational changes, and/or indirectly by activating their downstream signaling in a neurotrophin-independent mode [100]. Zinc and copper ions seem to induce restoring outcomes in several in vitro and in vivo models. Both metals rescued the level of NGF in zinc(II)-deficient mice, counteracted p75-driven apoptosis in the chick neural retina, and blocked the binding of NGF to p75, attenuating the triggered pro-apoptotic signaling cascade in chick embryo cell cultures [67]. Further, copper and zinc play a prominent role in the signaling during memory formation [87,101,102,103].

Accordingly, zinc treatment (micromolar concentrations) can boost BDNF mRNA expression in cortical cell cultures and promotes BDNF release and maturation [91,92,104]. A well-known pathway of synaptic plasticity. For these properties, zinc has been included in the integrative therapy for some psychiatric illnesses [105]. However, more generally, these metals can influence the pathways involved in CREB expression and phosphorylation [100,106,107], thus sustaining the multiple paths of CREB activity that have been shown to induce successful regeneration into the lesion site in spinal cord injury models [108,109].

The constant expansion in understanding the roles of zinc in normal human physiology is also accompanied by a new perspective on the role of zinc homeostasis management as a potential therapeutic tool in neuroregeneration. In the case of optic nerve damage, the death of retinal ganglion cells (RGCs) has been observed to occur in association with a rapid increase in mobile zinc in the retina, particularly in synaptic vesicles of amacrine cell processes, after optic nerve injury. In this study, by removing the Zn^2+^ transporter ZnT-3, or chelating the Zn^2+^ ions, the authors were successful in rescuing many injured CGRs and regenerating axons. This is a clear demonstration of the importance of Zn^2+^ dyshomeostasis in both RGC death and regenerative failure, and the clinical feasibility of zinc homeostasis management [110].

An increase in intracellular copper can modulate many proteins associated with the regulated secretory vesicle pathways, including synaptophysin, syntaxin-1, SNAP-25, and the LDCV proteins chromogranins A and C [111]. This agrees with the observations that copper internalization is associated with molecular steps essential in neuronal differentiation and neurite branching [112,113]. Remarkably, copper interplay with prion protein (PrPC) and binding within the physiological concentration range displays a functional role of PrPC in normal copper homeostasis in brain metabolism, whereas Cu^2+^ rapidly and reversibly stimulates the internalization of PrPC [114,115]. Hence, suggesting a copper contribution to PrPC-mediated survival signal and the promotion of neuritogenesis [116,117].

A large body of evidence associates metal dysregulation with motoneuron diseases [118,119]. Many studies have been focused on the role of aberrant metal homeostasis and binding in the aggregation process of SOD1 that has been shown to trigger SOD1 toxic deposition in amyotrophic lateral sclerosis (ALS) [83]. Furthermore, mutations in the copper transport gene ATP7A cause, among another two distinct human diseases, X-linked distal hereditary motor neuropathy, a condition that affects ATP7A intracellular trafficking and alters Cu levels within the nervous system. [84]. Based on the discovery that a CCS/G93A-SOD1 dual transgenic mice model of ALS develop accelerated neurological deficits, consistent with an apparent copper deficiency within the spinal cord, Williams et al. tested a therapeutic approach based on copper replenishment. Using the copper complex CuATSM, a safe vehicle of copper selective in cells with damaged mitochondria, they demonstrated that early CuATSM treatment of pups expressing SODG93A with CCS avoided the early mortality and allowed a normal grow [118]. According to this study, direct evidence reveals that altered metal uptake during specific early lifetime windows is associated with adult-onset ALS [85]. Thus, copper homeostasis is undeniably crucial in the maintenance and function of motor neurons, and therefore in the pathogenic mechanisms underlying motoneuron degeneration. This question deserves further examination with a view to possible connections with the functions of NTs.

Lithium is a metal with a long-lasting history of use as first-line drug for treating bipolar disorder, as well as recurrent depression [86]. Lithium-induced pharmacological effects were explained by replenishment with BDNF and NGF in patients with depression. Indeed, lithium treatment can upregulate expression and secretion of various NTs, including BDNF, NGF, and GDNF, as well as their receptors [79,80,89]. In addition, lithium’s effects are linked with its inhibitory activity on several cell signaling-related key enzymes, including glycogen synthase kinase-3β (GSK3β), inositol monophosphatase, phosphoadenosylphosphate phosphatase, and Akt/beta-arrestin 2 [93]. Nevertheless, accumulating evidence also attributes lithium’s properties to many other effects on neurotransmitter release/signaling, oxidative stress, apoptosis, hormonal modulation, upregulation, and secretion of neurotrophic factor, enhancement of neurogenesis and differentiation, learning and memory improvement [80,94,95,96,97,98].

These effects confirm the high potential of lithium to prevent neuronal degeneration [94] and to facilitate nerve regeneration, also enhancing the speed and extent of axons remyelination [90,94]. Conversely, the benefits of lithium treatment for bipolar disorders do not appear to be related to both NT-3 and NT-4/5 expression [88].

Moreover, lithium’s ability in reducing neuronal damage after acute spinal cord injury (SCI) by inducing autophagy is of great importance [120,121,122]. Lithium competition with magnesium (Mg^2+^) can depress the phosphoinositide cycle and leas to a net reduction in intracellular 1,4,5 inositol triphosphate (IP3), a known autophagy suppressor [123,124]. The overall mechanisms triggered by lithium make this metal a powerful and safe tool against many neuronal pathologies, where neuronal regeneration appears to be more challenging. Much evidence demonstrated that this avenue is open and waiting for more extensive applications for humans [125,126]. As a consequence of genetic background or environmental factors, deviation of metal homeostasis from the normal state may occur, ultimately resulting in functional changes in NT activities that can lead to an increased risk for disease onset and poor prognosis. Hence, interventions on the quality and quantity of metal–NT functional interplay can be prominent tools in the treatments of various neurological diseases that require neuroregenerative support.

## 7. Concluding Remarks and Future Perspectives

BDNF, GNDF, and NGF have long been considered to be the major regulators of motoneuron physiology and recovery from various insults. NTs play key roles in regenerative medicine because they are required not only for the restoration of trophic circuitry but also for triggering differentiation and axon targeting. However, NT activities are complex processes, and regulatory mechanisms of their expression, secretion, and binding to receptors need further understanding, also considering the more recent knowledge on metal participation in neurorecovery.

The progress in mechanistic investigations in the field of neurotrophin-mediated neuroprotection and neuroregeneration, aiming at a complete understanding of integrated pathways, offers possibilities for the development of novel neuroregenerative therapeutic approaches. This is an opportunity that warrants further attention.

## Figures and Tables

**Figure 1 biomolecules-12-01015-f001:**
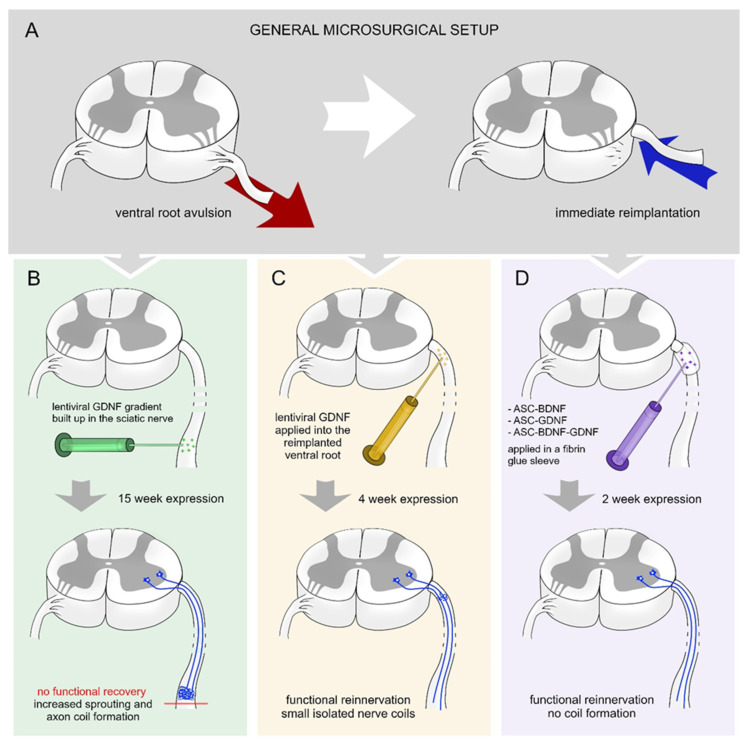
Various treatment strategies with BDNF and GDNF following ventral root injury. (**A**) Schematic overview of experimental ventral root avulsion and reimplantation. (**B**) Building up a proximo-distal gradient of GDNF [41] (15-week-long effect) in the sciatic nerve leads to robust axon coil formation that hinders the regeneration of the injured motoneurons and functional reinnervation of denervated hind limb muscles. (**C**) Shorter (4-week-long) expression period of time of GDNF [16] in the reimplanted ventral root results in the appearance of small isolated axon coils and functional recovery. (**D**) Transfected rat adipose tissue-derived stem cells (rASCs) grafted around the reimplanted ventral root producing GDNF and/or BDNF for 2 weeks [15] enhances elongative axon growth without coil formation and results in functional recovery.

**Figure 2 biomolecules-12-01015-f002:**
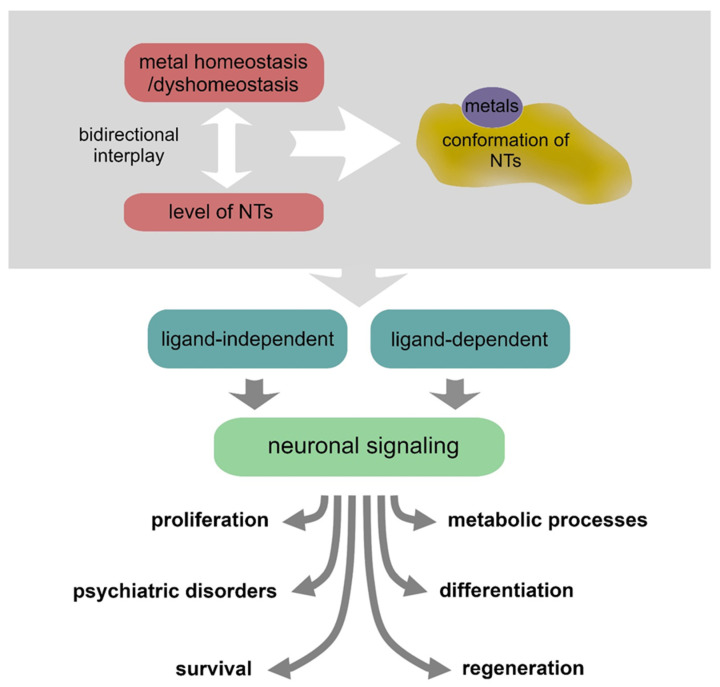
Metal homeostasis/dyshomeostasis can affect the expression, conformation and signaling of various neurotrophins, thus producing various effects on neuronal pathophysiology.

## Data Availability

Not applicable.

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
