# Peer review of "The Role of Metals in the Neuroregenerative Action of BDNF, GDNF, NGF and Other Neurotrophic Factors"

_biomolecules, 2022, doi:10.3390/biom12081015_

Round 1

Reviewer 1 Report

This review is particularly focused on roles of neurotrophins in motoneuron survival and recovery from injuries. The key role of metal homeostasis/dyshomeostasis and metal interactivity with neurotrophins on neuronal pathophysiology is also highlighted. This paper concludes a large number of recent researchwhich will help researchers to further develop this field. However, as the logic is not clear enough in the main body of this review, it still needs to be accepted after major revisions. My detailed comments are as follows: 

1. The division of the main frame is unreasonable. The main body of this paper is divided into six parts. Part 2, Part 3 and Part 4 introduce the role and molecular mechanism of BDNF and GDNF and their therapeutic potential in nerve regeneration. The fifth part introduces the interaction between neurotrophins and metals, and the sixth part introduces the role of metal ions in nerve regeneration. It is suggested that part 2, part 3 and part 4 be grouped together and integrate the narrative.

2. This paper lacks summary and outlook, resulting in the incomplete framework of this review. 

3. The logic of the main part is poor. For example, it is suggested that the third part be described in accordance with the advantages, disadvantages and improvement plans of neurotrophin treatment of sports injuries.

4. It is suggested that the contents of the fifth part be displayed in tabular form to facilitate readers to understand the interaction between metal ions and neurotrophins.

5. The authors are suggested to cite numbers of representative articles in the main text. For example, the article Receptor-targeting nanomaterials alleviate binge drinking-induced neurodegeneration as artificial neurotrophins (https://doi.org/10.1002/EXP.20210004) published in Exploration is very relevant to this review, which is recommended to be cited in the revision process.

Author Response

We thank the referee for evaluating our manuscript and suggesting ameliorations. Hence, we modified the text accordingly. Among the other many changes made to better harmonize the manuscript we did the following: minor issues were corrected; the division of the main frame has been modified; a summary has been added at the end; the section 4 (former 5) has been displayed also in tabular form to facilitate readers to understand the interaction between metal ions and neurotrophins; additional papers were cited and discussed for completeness (see NGF issue) and to better harmonize the review. But we are sorry, but we didn’t address the request for citation because the suggested paper is not clearly necessary to improve the quality of the manuscript under review.

Reviewer 2 Report

This is a very interesting review of the literature relating neurotrophic function and their modulation by metal ions. The manuscript is well written and is comprehensive of the literature.  A minor issue is that the review ends abruptly. Perhaps providing a succinct summary or discussing future clinical applications of the studies would add more of an impact to the review article. 

Minor additional issues:

Line 212- definition of GDNF is repeated here- is already defined earlier in the paper.

Lines 270-272 - This sentence is quite awkward.

Line 299 - the word "opened" should be "open".

Author Response

We thank the referee for evaluating our manuscript and suggesting ameliorations. Hence, we modified the text accordingly:

Among the other chages, minor issues were corrected accordingly.

A brief summary has been added at the end.

Reviewer 3 Report

In this review paper by Nicoletti et al entitled ’ Neurotrophins and metals in neuroregeneration’, the authors discussed about the neurotrophins, physiological significance of neurotrophins and their interaction with metals. Major focus is attributed to motoneurons and effect of neurotrophins is discussed in this aspect. The review is very well written and referenced. Some of the concerns are provided below

1. The title includes ‘neurotrophins…’ which is misleading since the authors only addressed BDNF and GDNF primarily in this manuscript providing separate sections for them. Neurotrophins is a much broader group of molecules with similar neurotrophic activity. The title must be changed according to the content of the manuscript.

2. Why did the authors focus only on motoneurons? This has not been explained anywhere in the text. Is it because GDNF and BDNF specially acts on motoneurons? It is understandable that the authors may have an area of expertise, but even when focusing on motoneurons – is it only that BDNF and GDNF modulates motoneurons function whereas other neurotrophins have no effect? Please explain these in the text.

3. The authors write in the abstract 'Neuroregenerative medicine can take great advantage by the deepening of knowledge of the molecular mechanisms underlying the neurotrophins properties'. This manuscript does not explain how (or which) ‘underlying pathways’ are implicated in neuroregeneration, nor how these pathways are modulated by metals in a pathological situation.

4. Page 3 line 109. The authors write ‘RET is mainly expressed in sensory neurons, dopaminergic….’ but reference for RET expression in motoneurons is missing!

5. In section 5 of this manuscript, the authors discuss the interaction of neurotrophins and metals. It seems that this section is out of place at the moment. In which way authors link it to the ‘motoneurons and neurotrophin’ issue? Any alteration in neurotrophin by metals will affects all the cells anyway. Are there specific circumstances when the effect on motoneurons become crucial? Moreover, this section is not focused towards motoneurons at all, although the authors describe in abstract that they want to focus on motoneurons in this manuscript!

6. Page 6 first paragraph – the authors states ‘..suggesting that any intervention on transition metals homeostasis can produce relevant effects…’. The authors should carefully discuss these since intervention with one or mix of metals may yield variable response from various types of neurons or other cell types in the body. Please focus and discuss the practically relevant issues.

7. NGF has been discussed in several places, but does not get an introduction like BDNF and GDNF, why?

8. Section 6: Some of the sentences are repetition of previous sections number 5!  This section as well is out of focus and spans from neural retina, cortical cell cultures, spinal cord injury, optic nerve,  retinal ganglion cells, etc etc.. but no motoneuron or related events. What is the point! Again, not all these cells behave similarly towards specific metals or neurotrophins!

9. In the closing statement the authors write - ‘Hence, interventions on the quality and quantity of metal/NTs functional interplay can be prominent tools in the treatments of various neurological diseases that require neuroregenerative support’. This statement seems vague. Was there any experimental evidence to support? Can you discuss how this can be achieved? How these can be prominent tool? to address which disease? How to specifically target it to specific type of cells?

Author Response

We thank the referee for evaluating our manuscript and suggesting ameliorations. Hence, we modified the text accordingly:

1: The title has been changed: (1 Neuroregenerative properties of GDNF, BDNF, NGF and other neurotrophins: a perspective on metals roles. -  2 GDNF, BDNF, NGF and other neurotrophins in neuroregeneration: metals roles.)

2: Motoneurons recovery represent a well-known example in the studies of neuroregeneration and GDNF, BDNF and NGF were shown to exert relevant role in the process. Other neurotrophins can have additional effect, but this issue warrants further understanding.

3: This review is aimed at suggesting new perspectives ad encouraging further studies on neuroregeneration to understand specific pathways and targeting that can open the way to valuable therapeutic tools. To this purpose, in the last two sections, we described different underlying pathways modulated by metals as possible mechanisms of neuroprotection. See some sentences as example in lines: 189-191; 192-198; 203-208. In addition, a possible mechanisms that involve copper homeostasis and its therapeutic potential in ALS has been added in the last section.

4: A specific reference has been added

5: In the abstract we state: “This review is particularly focused on the roles of neurotrophins in motoneuron survival and recovery from injuries and evaluates the therapeutic potential of neurotrophins. The key role of metal homeostasis/dyshomeostasis and metal interactivity with neurotrophins on neuronal pathophysiology is also highlighted.” These two aspects are treated in separate sections because the metal importance for neuron pathophysiology is to date mostly studied in the field of neurodegeneration. The underlining molecular events, however, could be of interest for motor neuron recovery as well, and this review could stimulate extended studies on neurotrophins/metals role. However, we added some sentences to better connect the section with motorneuron issue.

6: The first paragraph at page 6 is actually the last of section 5, and the sentence ‘…suggesting that any intervention on transition metals homeostasis can produce relevant effects…’ is intended as a warning in the use of metals or metal homeostasis modulators due to the various, still not fully understood, effects. In fact, the paragraph ends with the sentence: “This issue is still debated but could deserve deeper understanding to unveil the molecular basis of neural recovery.”

In addition, the sentence opens also to the next section that describes the therapeutic potentials of metal-based interventions.

However, to ameliorate this part, the referee’s suggestion has been accepted and the text has been amended accordingly.

7: NGF importance has been also described.

8: Sometime a sentence repetition could be useful to highlight an issue from a different perspective. However, if the author had been more precise we would have answered more adequately.

The metals role in the physiology of neurotrophins is a relatively recent issue and the sections are aimed at collecting as much information related to the various mechanisms and possible target of intervention. This obviously highlighted also different cell-specific outcome that we consider important as well, particularly to stimulate further studies and potential motorneuron specific applications.

Round 2

Reviewer 3 Report

The author's have incorporated several changes as suggested previously, but the overall quality of this manuscript falls short from reaching publishable quality.

One of the prime reason being this review manuscript is primarily driven by speculation. Thought provoking reviews are a great way of conveying innovative ideas and future prospects among the field, but only when solid background evidence are provided, which is lacking in this manuscript.

Author Response

We thank the referee for the manuscript evaluation and criticisms. We further revised our manuscript after Editor's suggestions and hope that now could be acceptable for publication.